# Implementation and Evaluation of a Wide-Range Human-Sensing System Based on Cooperating Multiple Range Image Sensors

**DOI:** 10.3390/s19051172

**Published:** 2019-03-07

**Authors:** Mikihiro Tokuoka, Naoki Komiya, Hiroshi Mizoguchi, Ryohei Egusa, Shigenori Inagaki, Fusako Kusunoki

**Affiliations:** 1Tokyo University of Science, 2641, Yamazaki, Noda, Chiba 278-0022, Japan; 7518523@ed.tus.ac.jp (N.K.); hm@rs.noda.tus.ac.jp (H.M.); 2Meiji Gakuin University, 1-2-37, Shirokanedai, Minato-ku, Tokyo 108-0071, Japan; egusa@psy.meijigakuin.ac.jp; 3Kobe University, 3-11, Tsurukabuto, Nada, Kobe, Hyogo 657-8501, Japan; inagakis@kobe-u.ac.jp; 4Tama Art University, 2-1723, Yarimizu, Hachioji, Tokyo 192-0375, Japan; kusunoki@tamabi.ac.jp

**Keywords:** three-dimensional range image sensor, time synchronization, coordinate transformation matrix, sensing interest, learning support system

## Abstract

A museum is an important place for science education for children. The learning method in the museum is reading exhibits and explanations. Museums are investing efforts to quantify interests using questionnaires and sensors to improve their exhibitions and explanations. Therefore, even in places where many people gather, such as in museums, it is necessary to quantify people’s interest by sensing behavior of multiple people. However, this has not yet been realized. We aim to quantify the interest by sensing a wide range of human behavior for multiple people by coordinating multiple noncontact sensors. When coordinating multiple sensors, the coordinates and the time of each sensor differ. To solve these problems, coordinates were transformed using a simultaneous transformation matrix and time synchronization was performed using unified time. The effectiveness of this proposal was verified through experimental evaluation. Furthermore, we evaluated the actual museum content. In this paper, we describe the proposed method and the results of the evaluation experiment.

## 1. Introduction

The museum is very important as a place for science education for children [1]. This is because the museum helps children gain knowledge by learning and through experience using content and learning materials [2]. The learning material and content in the museum are mainly about the exhibit and their explanations. Moreover, a panel and video tape recorder of explanation is used to complement the exhibits. In recent years, proposals have been drawn for further improving learning using these contents and learning materials. This proposal discriminates popular and unpopular exhibits using questionnaires and interviews, and changes the exhibits from time to time [3]. This is very important in further improving the efficiency of learning. In addition, research and development of museum learning support systems in science education is being researched and developed as a way to support children’s learning [4,5,6,7]. As a method to evaluate these systems, subjective evaluations such as questionnaires and interviews, which are similar to evaluations of exhibits, are used frequently [4,5,6,7,8,9]. However, there are major problems in these evaluation methods; it is inefficient because it is time consuming. Furthermore, because it is a formal structured mechanism, it is difficult to obtain the natural opinion of the learner. Because we conduct interviews and questionnaires after the children have experienced all content, we cannot quantitatively calculate the interest at a certain exhibit at a given time. Therefore, in essence, we cannot improve the content or learning materials. This is a problem that needs to be solved.

To solve these problems, techniques for sensing the interests of people have attracted attention and various research studies have been carried out. Some studies use a contact/noncontact sensor; studies that use contact sensors include measurements of physiological phenomena, electro dermal activity (EDA) [10] and electroencephalograms [11,12], estimation of a line of sight using eye gaze capturing devices, and blinking related to the most interests [13,14]. However, although these can be quantitatively measured, there is a problem that natural opinions cannot be obtained as the subjects may be stressed, and it takes time to wear these devices. Meanwhile, research using noncontact sensors includes measuring blinks using a web camera [15,16,17,18,19,20] and measuring the line-of-sight using an installation-type measuring instrument such as Tobii [21]. However, since these are noncontact methods, which allow measuring the natural body of the experiencer, there is a problem in that the measurement range is limited. For example, in a situation where there are many people like in a museum, there are cases where people cannot be tracked (Figure 1). Because people may overlap each other in the video, the sensor cannot recognize more than one person at a time. Therefore, it is necessary to measure these parameters within a range of 5–10 m. Therefore, there is a need for a system that measures quantitatively and in a noncontact manner the interest of “the place and the time” in a wide range. The learning effect in an actual museum has not been evaluated using our proposed technique.

We propose a system to solve these problems. By cooperating multiple noncontact sensors, quantitative interest in a wide range can be estimated in a noncontact manner. In particular, the proposed system observes behaviors such as a line-of-sight and eye blink for multiple learners at all times by using a large number of cooperating sensor groups arranged in the environment (for example, in the museum). 

In this paper, we describe a method of coordinate transformation and time synchronization that can enable cooperation of multiple sensors and the results of the evaluation experiment. In addition, we describe the results of evaluating the contents and learning materials implemented in actual museums.

## 2. System

### 2.1. System Overview

We developed a system to estimate quantitative interests in a wide range by coordinating multiple noncontact sensors. The proposed system constantly observes the behavior of a learner by using a large number of sensor groups arranged in a certain environment. Accordingly, we observed the interests of learners. Based on the obtained results, we realized the quantification of the interest “the place and the time” which could only be imagined fragmentarily through interviews and questionnaire-based surveys.

Figure 2 shows a model image of the system, and Figure 3 shows the system setup. The proposed system consists of a sensor group comprising multiple noncontact sensors and a data storage unit that accumulates all acquired data. The data storage unit includes elements, such as the direction of the learner’s face, detection of blinking, and gaze time in the gaze direction, related to the learner’s interest. In this paper, as the first step in realizing the system, we measure eye blinks, which are said to be most affected by human interest [22,23,24].

To realize the system, we used a Kinect sensor [25] in this study. Microsoft’s Kinect sensor is a range-image sensor originally developed as part of an indoor video-gaming system. Although it is inexpensive, the sensor can obtain sophisticated measurements and adjudge the user’s location. In addition, this sensor can recognize humans and the human skeleton using the library in Kinect’s software development kit for Windows. The Kinect senor can measure a three-dimensional skeletal location composed of 25 points on the human body, including the hands and the legs, and it can identify the user’s pose or status based on these functions. This skeletal information makes it possible to recognize various body movements. 

Coordinate transformation and time synchronization for cooperation of multiple sensors are required for enlarging the measurement range by cooperating with multiple Kinect sensors. Coordinate transformation, time synchronization, and blink detection are all automatically initiated on our developed program. We describe our proposed methods below.

### 2.2. Coordinate Transformation Using Simultaneous Transformation Matrix

#### 2.2.1. Coordinate Transformation

When the area requiring measurement is wide or there is a considerable amount of information to be measured, it is necessary to expand the measurement range by cooperating multiple sensors. However, when trying to cooperate multiple position measuring sensors, the coordinate system of each sensor and the measured value should be independent. As shown in Figure 4, the coordinate system of sensors 1, 2, and 3, and the measured value are independent. To measure the same object, it is necessary to unify it to one coordinate system, as shown in Figure 5. 

In general, in many position measurement sensors such as the Kinect sensor, the measurement results in the coordinate system remain unique to each sensor. Therefore, simply by performing arbitrary multiple sensor placement and measurement, even if the same target position is measured, the output values of the sensors do not match and it is very difficult to coordinate the measured values. Various studies have been conducted to solve this problem. Unfortunately, conventional research also failed to accurately realize coordinate transformation through the cooperation of multiple sensors [26]. As the main method, those studies used a checker board for calibration [27]. However, although this method can unify the exact coordinate system, it takes a considerable amount of time to unify the coordinate system. 

It takes a few hours for the checkerboard to move several centimeters at a time [27]. It is difficult to implement this in an environment such as a museum, where this has to be realized ad hoc.

We therefore propose a method of using the simultaneous coordinate transformation matrix to unify all coordinate systems in an arbitrary coordinate system. This is a method that can perform coordinate transformation in a short time and coordinate unification.

#### 2.2.2. Method of Coordinate Transformation Using Simultaneous Transformation Matrix

We describe coordinate transformation using a simultaneous transformation matrix. We define P(x, y, z) as the coordinates as seen from the coordinate system unique to Kinect sensor at the point P in space. Furthermore, we define P′(x′, y′, z′) as the coordinates in the unified coordinate system at the point P in the space. P is the coordinate system of Figure 4, P′ is the coordinate system of Figure 5 after unification. At this time, by using the coordinate transformation matrix T, it is possible to convert to P′ from P, using Equations (1) and (2).
(1)$$\left(\begin{array}{c} x^{\prime }\\ y^{\prime }\\ z^{\prime }\\ 1 \end{array}\right)=\left(\begin{array}{cccc} r_{11} & r_{12} & r_{13} & q_{x}\\ r_{21} & r_{22} & r_{23} & q_{y}\\ r_{31} & r_{32} & r_{33} & q_{z}\\ 0 & 0 & 0 & 1 \end{array}\right)\left(\begin{array}{c} x\\ y\\ z\\ 1 \end{array}\right),$$
(2)$$P^{\prime }=TP,$$

Coordinate representation such as that in Equation (2) is called a simultaneous coordinate representation. This can express the movement of the coordinate system by the expression of multiplication of one coordinate transformation matrix. We describe each component of the coordinate transformation matrix T as follows. Figure 6 shows the component of T.

In Figure 6, the components from r_11_ to r_33_ represent the rotational movement of the coordinate system, and they are expressed by the multiplication of the equation of the rotational movement around each axis.
(3)$$\left(\begin{array}{ccc} r_{11} & r_{12} & r_{13}\\ r_{21} & r_{22} & r_{23}\\ r_{31} & r_{32} & r_{33} \end{array}\right)=\;R_{x}R_{y}R_{z},$$

We describe the rotational movement R_x_ in the coordinate system around the X axis as an example of the rotational movement around the axis. As shown in Figure 7, when the coordinate system is rotated by θ_x_ about the X axis, the coordinates P′(x′, y′, z′) of the point in the coordinate system after movement are given by Equation (4). Equation (4) is expressed using coordinates P(x, y, z) and θ_x_.
(4)$$\begin{array}{c} x^{\prime }=x,\\ y^{\prime }=ycos\theta_{x}+zsin\theta_{x},\\ z^{\prime }=-ysin\theta_{x}+zcos\theta_{x}, \end{array}$$

When Equation (4) is expressed as a matrix, Equation (5) is expressed as
(5)$$\left(\begin{array}{c} x^{\prime }\\ y^{\prime }\\ z^{\prime } \end{array}\right)=\left(\begin{array}{ccc} 1 & 0 & 0\\ 0 & cos\theta_{x} & sin\theta_{x}\\ 0 & -sin\theta_{x} & cos\theta_{x} \end{array}\right)\left(\begin{array}{c} x\\ y\\ z \end{array}\right),$$

From Equation (5), the rotational movement around the X axis is represented by a matrix of Equation (6).
(6)$$R_{x}=\left(\begin{array}{ccc} 1 & 0 & 0\\ 0 & cos\theta_{x} & sin\theta_{x}\\ 0 & -sin\theta_{x} & cos\theta_{x} \end{array}\right),$$

Similarly, in the case of rotation about the Y and Z axes, the expressions for converting the coordinates on the original coordinate system to the coordinates in the coordinate system after rotation are expressed by Equations (7) and (8).
(7)$$R_{y}=\left(\begin{array}{ccc} cos\theta_{y} & 0 & -sin\theta_{y}\\ 0 & 1 & 0\\ sin\theta_{y} & 0 & cos\theta_{y} \end{array}\right),$$
(8)$$R_{z}=\left(\begin{array}{ccc} cos\theta_{z} & sin\theta_{z} & 0\\ -sin\theta_{z} & cos\theta_{z} & 0\\ 0 & 0 & 1 \end{array}\right),$$

Substituting Equations (7) and (8) into Equation (3) gets
$$\left(\begin{array}{ccc} r_{11} & r_{12} & r_{13}\\ r_{21} & r_{22} & r_{23}\\ r_{31} & r_{32} & r_{33} \end{array}\right)=\;R_{x}R_{y}R_{z}$$
(9)$$=\left(\begin{array}{ccc} cos\theta_{y}cos\theta_{z} & cos\theta_{y}sin\theta_{z} & -sin\theta_{y}\\ sin\theta_{x}sin\theta_{y}cos\theta_{z}-cos\theta_{x}sin\theta_{z} & sin\theta_{x}sin\theta_{y}sin\theta_{z}+cos\theta_{x}cos\theta_{z} & sin\theta_{x}cos\theta_{y}\\ cos\theta_{x}sin\theta_{y}cos\theta_{z}+sin\theta_{x}sin\theta_{z} & cos\theta_{x}sin\theta_{y}sin\theta_{z}-sin\theta_{x}cos\theta_{z} & cos\theta_{x}cos\theta_{y} \end{array}\right),$$

Next, q_x_, q_y_, q_z_ are components representing parallel movement in each axis direction. Combining the rotational movement and the parallel movement described above, the movement of the coordinates is expressed by
(10)$$\left(\begin{array}{c} x^{\prime }\\ y^{\prime }\\ z^{\prime } \end{array}\right)=\left(\begin{array}{ccc} r_{11} & r_{12} & r_{13}\\ r_{21} & r_{22} & r_{23}\\ r_{31} & r_{32} & r_{33} \end{array}\right)\left(\begin{array}{c} x\\ y\\ z \end{array}\right)+\;\left(\begin{array}{c} q_{x}\\ q_{y}\\ q_{z} \end{array}\right),$$

Equation (10) can thus be expressed as
(11)$$\left(\begin{array}{c} x^{\prime }\\ y^{\prime }\\ z^{\prime } \end{array}\right)\;=\left(\begin{array}{ccc} r_{11} & r_{12} & r_{13}\\ r_{21} & r_{22} & r_{23}\\ r_{31} & r_{32} & r_{33} \end{array}\right)\left(\begin{array}{c} x\\ y\\ z \end{array}\right)+\left(\begin{array}{c} q_{x}\\ q_{y}\\ q_{z} \end{array}\right)=\left(\begin{array}{cccc} r_{11} & r_{12} & r_{13} & q_{x}\\ r_{21} & r_{22} & r_{23} & q_{y}\\ r_{31} & r_{32} & r_{33} & q_{z} \end{array}\right)\left(\begin{array}{c} x\\ y\\ z\\ 1 \end{array}\right),$$

Finally, the fourth line in Figure 6 represents the scaling of the coordinate system. In the case of the coordinate transformation used in this research, it is no need to enlarge/reduce; therefore, it is set to 1 (equal magnification), as shown in Equation (12).
(12)$$1=\left(0\;0\;0\;1\right)\left(\begin{array}{c} x\\ y\\ z\\ 1 \end{array}\right),$$

Next, we describe how to calculate such a coordinate transformation matrix T. T can be calculated from the correspondence relationship between both coordinate systems if there is a point where the coordinates seen from the coordinate system of both the Kinect’s coordinate system and the unified coordinate system are known. 

When there are *n* points in space, as shown in Figure 8, the coordinates in the Kinect coordinate system of the *n*th point are defined as PnS(xnS,ynS,znS) and those in the unified coordinate system as PnF(xnF,ynF,znF). When the coordinates in the homogeneous coordinate system are represented by a matrix, the coordinates of the 1st to nth points as viewed from each coordinate system are expressed by
(13)PS=(x1Sx2S⋯xnSy1Sy2S⋯ynSz1Sz2S⋯znS11⋯1) PF=(x1Fx2F⋯xnFy1Fy2F⋯ynFz1Fz2F⋯znF11⋯1),

Therefore, letting TFS be the coordinate transformation matrix for transforming the coordinates of the points in the Kinect coordinate system into the coordinates in the unified coordinate system, this coordinate transformation can be expressed as
(14)(x1Fx2F⋯xnFy1Fy2F⋯ynFz1Fz2F⋯znF11⋯1)=TFS (x1Sx2S⋯xnSy1Sy2S⋯ynSz1Sz2S⋯znS11⋯1)
(15)PF=TFSPS,

From Equation (15), a coordinate transformation matrix TFS is calculated. When Equation (14) is n = 4 and these points are not on the same plane, the coordinate transformation matrix TFS can be obtained as follows, using the inverse matrix of PS [28].
(16)TFS=PFP−1S,

We calculate the coordinate transformation matrix by measuring with n = 4. To unify Kinect’s coordinate system by this method, the point where unified coordinates are known on the real space and can be measured by Kinect (hereinafter, such a point is called a sample point) should be set. Therefore, in this research, the measurement result of the sample point measured by one Kinect (hereinafter referred to as the origin Kinect) is treated as a true value. Then, by substituting the true value and the results measured by other Kinect into Equations (14)–(16), the coordinate system of each sensor can be unified to the coordinates of the origin Kinect. After the calculation of the transformation matrix, each sensor is multiplied by a transformation matrix as shown below for the three-dimensional coordinate measurement result measured by other Kinect Sensor.
(17)(xFyFzF1)=(r11r12r13−qxr21r22r23−qyr31r32r33−qz0001)(xSySzS1),
(18)xF= r11xS+r12yS+r13zS,yF= r21xS+r22yS+r23zS,zF= r31xS+r32yS+r33zS,

Through these calculations, it becomes possible to treat the three-dimensional coordinate measurement result measured by all Kinects as a unified value in the coordinate system of the origin Kinect. All these matrix calculations are performed on a program.

### 2.3. Time Synchronization

#### 2.3.1. Summary

In the measurement by multiple sensors, the measurement start timing and the end timing are different because the time of each sensor is different. As shown in Figure 9, even if the same operation is recognized, a time lag occurs, and an accurate analysis cannot be performed; therefore, time synchronization of multiple sensors is indispensable. By performing time synchronization as shown in Figure 10, it is possible to perform an accurate analysis using multiple sensors. Nevertheless, with the conventional synchronization of multiple Kinect sensors, time synchronization has not been successful [29,30]. Conventional time-synchronization research is based on the establishment of a unified time and server. Previous studies that used a unified time did not state about the accuracy of time synchronization, assuming that the cycle of unified time does not fluctuate. However, there is a possibility that the unified time deviates. In addition, any research that establishes a server and synchronizes time is accurate. However, to process and synchronize information with high capacity and geometric qualities like the Kinect V2 sensor with color images and depth information, it is necessary to establish an expensive server with a fast processing time. The establishment of an expensive server has no versatility if we consider the realization at our museum; this is the original purpose of our study. Therefore, we did not set up a server, and referenced the existing server, conducted time synchronization by considering the processing time of each personal computer, and evaluated the accuracy.

#### 2.3.2. Time Synchronization Using Unified Clock

First, we created unified time and recorded the unified time on multiple sensors; then, we unified the time of multiple sensors to this unified time and realized time synchronization. Figure 11 shows the occurrence of this time synchronization. After the measurement, we unified the recorded time of all the sensors to the unified time. The frequency of the Kinect was 30 Hz, and the clock for the unified time was 100 Hz. By using this method, we ensured accurate time synchronization.

Next, we explain the algorithm of time synchronization. A personal computer was used to connect Kinect Sensors 1 and 2 and record the unified time on the Internet by using the DataTime class in the program. Time on the Internet refers directly to the time of the existing NTP server, and in this research, we referred to the NTP server of Tokyo Science University (nodarntp.rs.noda.tus.ac.jp); it can be used within our university. The time of the Internet was recorded at 30 fps, which is the sampling rate of a Kinect Sensor. However, depending on the performance of the personal computer, the recorded time differs from the unified time by approximately 1 ms. Therefore, the error between the times of the NTP server and recording was constantly calculated in each personal computer, and a correction was made. As a result, time synchronization was performed.

### 2.4. Eye Blink Detection

It is said that people’s interest can be gauged by the blinking of their eyes. We focused on the eye blink to quantify the interest of multiple people at the museum. Clearly, eye blinking is suppressed when an entity has caught people’s interests. In other words, when people are more interested, the number of their eye blinks decreases. If we measure eye blink at all times, we quantitatively estimate the interest, engagement, and excitement “on the spot, at the time”. However, in the conventional eye-blink measurement method, the measurement range is narrow because when learners look away, part of the eye-blink data is missed. The realization of the proposed wide-range eye-blink-measurement method could capture the eye-blink even when learners look away because its measures eye blinks using a coordinated sensor. In other words, the sensor always measures and records eye blinks. By measuring interest quantitatively, we clarified the learners’ interest in the contents of the museums. 

Eye-blink detection follows the flow of human detection, skeleton information detection, face recognition, and finally pupil detection, as detailed in the following text. First, the Kinect sensor performs human detection by using a database based on the learning data of skeletal information of a large number of people. Next, the sensor identifies a person based on the person’s coordinate information. Accordingly, the same person was tracked in a multipeople environment. Then, the sensor recognizes a human face and extracts the 3D coordinates of 1347 points on the face as feature points [31]. Based on these information, the eye position, and then the eye blinking were determined. The two states of the pupil (open/closed eyes) were determined based on the ratio of the iris width to the maximum iris width obtained by counting the number of black pixels. The state was recognized as “OPEN” when opening eyes and “CLOSE” when closing eyes based on the set threshold value. It is said that people’s interest can be gauged using their eye blink. Figure 12 shows the appearance of OPEN and CLOSE eyes. As shown in Figure 12, when the state changes from OPEN to CLOSE, and to OPEN again, we count it as one eye blink; the measurement cycle was 30 Hz.

In this way, the number of blinks is automatically measured.

## 3. Experiments

We describe the evaluation experiments and the results to evaluate the effectiveness of the coordinate transformation and time synchronization proposal method for multiple sensor cooperation, which is necessary for expanding the measurement range. In addition, we describe the results of evaluating the content of the museum using the proposed system.

### 3.1. Coordinate Transformation

We conducted an evaluation experiment on whether the coordinate system can be unified by coordinate transformation using simultaneous transformation matrix. As the first step in realizing the system, coordinate conversion was carried out using two Kinect Sensors to evaluate the coordinate system unification.

#### 3.1.1. Evaluation Experiment of Coordinate Transformation

In this experiment, a jig with four sampling points that are not on the same plane for the unification of two Kinect sensors and coordinate system was used [28]. Figure 13 and Figure 14 show the experiment setup. We unify the coordinate system of Kinect Sensor 1 and Kinect Sensor 2, as shown in Figure 13. First, as shown in Figure 13, a jig with sampling points is placed in front of two Kinect sensors, which measure the four-point coordinates of the jig. The simultaneous transformation matrix is calculated based on the coordinates of the four points measured by each Kinetic sensor. Next, one subject stands at a fixed measurement point, and each Kinect sensors measures the subject’s coordinates simultaneously. The measurement result is coordinate-transformed by the calculated coordinate transformation matrix to unify the coordinate system. The total number of measurement points is 66 points. Figure 15 shows the measurement points. Next, we describe the method of processing persons and images programmed. First, each Kinect sensor recognizes the coordinates of sampling points for calculating the coordinate transformation matrix. Infrared rays were then irradiated into the field-of-view of the sensor and the position coordinates of the specified sampling point were measured according to the reflection time, depth information, and color-image information. Based on the coordinates measured using Kinect sensors 1 and 2, the matrix conversion from matrix to expression was performed on all the PCs connected to the two Kinect sensors on the program to calculate the coordinate transformation matrix. Based on the calculated coordinate transformation matrix, the coordinate system of Kinect sensor 2 was unified to the coordinate system of Kinect sensor 1. Therefore, it is possible to link all the information on the coordinate axis of Kinect sensor 1.

#### 3.1.2. Evaluation Experiment Result of Coordinate Transformation

We evaluate the error of the coordinate measurement result of Kinect Sensor 1 and the coordinate result obtained by coordinate conversion of the measurement value of Kinect Sensor 2 using the coordinate transformation matrix. 

Figure 16 shows the coordinates of each measurement point after coordinate transformation; this is the measurement result of Kinect Sensor 1 and the measurement result of Kinect Sensor 2 that unified the coordinate system using coordinate transformation. In Figure 15, points that could not be measured outside the recognition range of the Kinect sensor are not plotted. From this result, it can be seen that the coordinate transformation is performed by the simultaneous transformation matrix. Next, we evaluate the error of coordinate transformation. The error is the difference in distance between the known coordinates and the distance difference between the known coordinates and the coordinates of person coordinates measured by Kinect Sensor 2; this is the distance between the known coordinates of the X-Z plane and the coordinates after coordinate system unification.

The museum content we evaluate is for children. It is necessary to distinguish between two or more children, and therefore, we set the allowable error as half of the shoulder width of the child. Because the average shoulder width of the child is 33.86 cm, the allowable error is 16.93 cm [32]. Table 1 summarizes the errors of the coordinates obtained by coordinate conversion of the coordinates measured by Kinect sensor 2 in this experiment. All experimental results are within tolerance. The average error value of the coordinate transformation was 4.18 [cm], and the standard deviation was ±2.98 [cm]. Thus, coordinate transformation by using the Kinect sensor can unify the coordinate system with an accuracy of 4.18 ± 2.98 [cm]. Therefore, the usefulness of the proposed method is proved.

### 3.2. Evaluation Experiment on Time Synchronization

We conducted an evaluation experiment on whether the time synchronization of two Kinect sensors can be achieved using time synchronization with a clock for unified time.

#### 3.2.1. Evaluation Experiment of Range of Measurement

In this experiment, we used a clock to unify time using two Kinect sensors. Figure 17 shows the experiment environment. We performed time synchronization between Kinect sensors 1 and 2, which are shown in Figure 17. We explain the experimental procedure below. First, we start measuring using Kinect sensor 1. Second, we start measuring Kinect sensor 2. Therefore, both Kinect sensors measure at and record for different arbitrary times. The measuring time is 53 s for each Kinect. After the end of the measurement, we standardized the Kinect sensors 1 and 2 with a clock for unified time. Through these experiments, we evaluate whether it is possible to synchronize time between Kinect sensors 1 and 2.

#### 3.2.2. Experimental Result

The results of the verification experiment are shown below. Table 2 shows the experiment start time, experiment end time, and elapsed time for each of the unified time. Unified time started at 0 s and ended at 53 s. We evaluate the elapsed time of the unified time and Kinect sensors 1 and 2. 53 s have elapsed in unified time, 52.954 s have elapsed in Kinect 1, and 53.033 s have elapsed in Kinect 2. Therefore, the difference between the elapsed times of Kinect sensors 1 and 2 was 0.079 s. The average value of time-synchronization error was 0.0069 [s], and the standard deviation was ±0.0024 [s]. As a result, time synchronization using the NTP server can be synchronized with an accuracy of 0.0069 ± 0.0024 [s]. Furthermore, we must accurately detect the eye blink, which occurs once, averaging 0.2 s [33]; therefore, we set 0.2 s as the allowable error. Since the difference in elapsed time is within the allowable error, the effectiveness of the proposed method for time synchronization was suggested.

### 3.3. Range of Measurement

We conduct an evaluation experiment on measure the range expansion enabled by linking two Kinect Sensors.

#### 3.3.1. Evaluation Experiment of Range of Measurement

In this experiment, we used two Kinect Sensors and one subject. Figure 18 shows the state of the experiment. In the experiment, the subject walked in front of the two Kinect Sensors. The walking route is as shown in Figure 18. At that time, a comparison is made between the area of one Kinect Sensor person within the discovery range and the area of two Kinect Sensors person within the discovery ranges.

#### 3.3.2. Evaluation Experiment Result of Range of Measurement

We describe the experimental results. Figure 19a shows a person discovery range with one Kinect sensor. Figure 19b shows a person discovery range with two Kinect Sensors. The person tracking area using one Kinect Sensor is 7.028 m^2^, and that using two Kinect Sensor is 11.23 m^2^. The area was calculated using an approximation curve. From this result, it was found that the measurement area can be expanded by cooperating multiple Kinect sensors. By cooperating two Kinect sensors, it is possible to trace a person in the range of 5 m × 4 m. It was suggested that it is possible to further extend the measurement range by increasing the number of Kinect sensors.

### 3.4. Evaluation Experiment of Contents

From Section 3.1, Section 3.2 and Section 3.3, we conducted evaluation experiment on the elemental technologies to realize the proposed system. We evaluate contents (multiple movies) implemented in an actual museum by the proposed system using these element technologies. This museum learning support system includes what I develop [34].

#### 3.4.1. Experimental Method

In this experiment, we evaluate contents and learning materials implemented in actual museums. We sensed four learners watching actual videos flowing in the museum using two connected Kinect sensors. During the experiment, we evaluate whether contents could be evaluated by an eye blink. The video is composed of different contents of four sections.

Figure 20 shows the experimental environment. By cooperating with two Kinect sensors that became clear in our experiment, it is possible for a person to track a range of 5 m × 4 m. Therefore, the museum contents to be evaluated in this experiment is done within 5 m × 4 m. We sense the interests of the four people who experience the content within this range.

#### 3.4.2. Experimental Result

We describe the experimental results of one subject as an example. One eye blink is converted on the graph as shown in Figure 21. Conversion result are shown in Figure 21. In sections 1, 3, 5, and 7, subjects are taking a break. In sections 2, 4, 6, and 8, subjects are viewing content videos of museums. First, during the experiment, it is understood that the data is always acquired without data loss. As shown in Figure 22, there are many eye blinks during a break. We evaluate based on this result. First, the eye blink rate is calculated as
(19)$$\left(\mathrm{Eye}\;\mathrm{blink}\;\mathrm{rate}\right)\;\left[\mathrm{times}/\min\right]=\frac{\left(\mathrm{Number}\;\mathrm{of}\;\mathrm{Eye}\;\mathrm{Blink}\right)\;\left[\mathrm{times}\right]}{\left(\mathrm{Elapsed}\;\mathrm{Time}\right)\;\left[s\right]}\times 60,$$

The results of calculating the blink rate are shown in Table 3. It is said that eye blink is suppressed when people has interest or attention [35]. In other words, the more subject have interest, the smaller subject’s eye blink rate is. As shown in Table 3, eye blink rate increases during a break. In addition, when subjects watched content video, eye blink rate decreases and eye blink is suppressed. From this, it is confirmed that the subject is interested when subject is watching the content video. Next, the degree of interest is described. Eye blink rate is high in the order of 4, 8, 2, 6 from Table 3. From this, it turns out that the interest is high in the order of 4, 8, 2, 6. This result is supported by a questionnaire to the subjects. From these results, the effectiveness of the proposed method was suggested. It was possible to evaluate the content in museum by quantifying the degree of interest of the learner by the proposed method.

## 4. Conclusions

In this paper, we described a method of coordinate transformation and time synchronization for the coordination of multiple sensors required for enlarging the measurement range, and discussed the evaluation results. To expand the measurement range by using multiple Kinect sensors, we proposed coordinate transformation using a simultaneous transformation matrix, ad hoc coordinate transformation, and time synchronization using unified time. In the experiment, we conducted experiments to evaluate the coordinate transformation using simultaneous transformation matrices and the usefulness of time synchronization using unified time. As a result, coordinate transformation was realized with accuracy, and it can identify multiple children. Furthermore, time synchronization could be realized with accuracy of detecting eye blinks. These results showed the effectiveness of the proposed method.

In the future work, we aim to add attention time or action in face orientation as an element to sense learners. By doing this, we aim to further quantify learner’s interests.

## Figures and Tables

**Figure 1 sensors-19-01172-f001:**
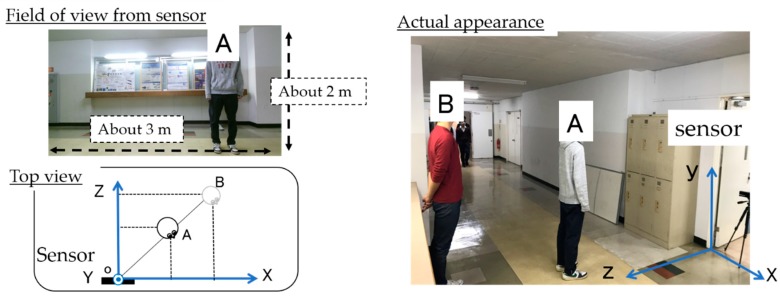
When people overlap each other in the sensor’s line of sight, the sensor cannot track these people. The sensor cannot recognize person B hidden behind person A.

**Figure 2 sensors-19-01172-f002:**
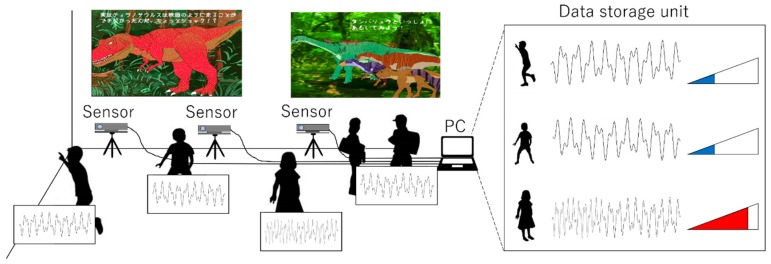
Model image of the proposed system. Simultaneously sensing the interests of multiple learners.

**Figure 3 sensors-19-01172-f003:**
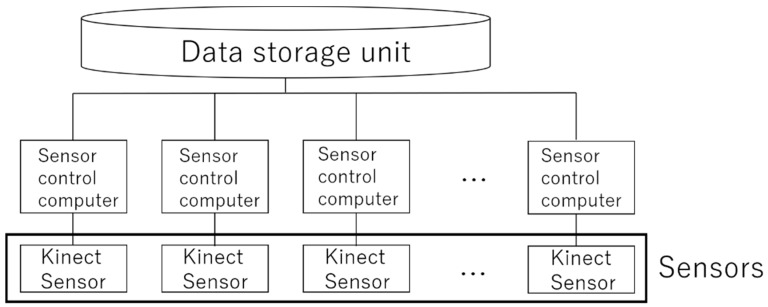
Setup of the proposed system.

**Figure 4 sensors-19-01172-f004:**
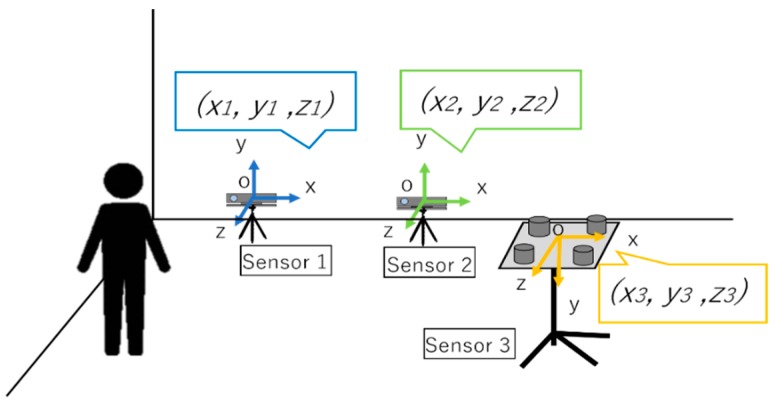
Issues in using multiple sensors. Even if the same target position is measured, the output values of the sensors do not match.

**Figure 5 sensors-19-01172-f005:**
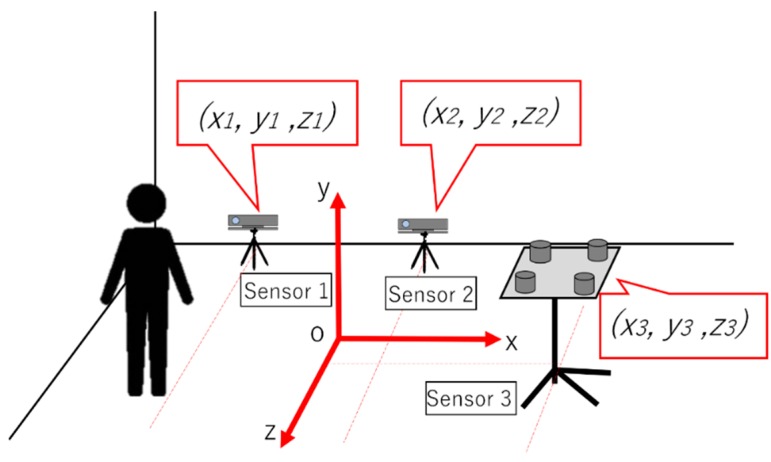
Unification of coordinate system.

**Figure 6 sensors-19-01172-f006:**
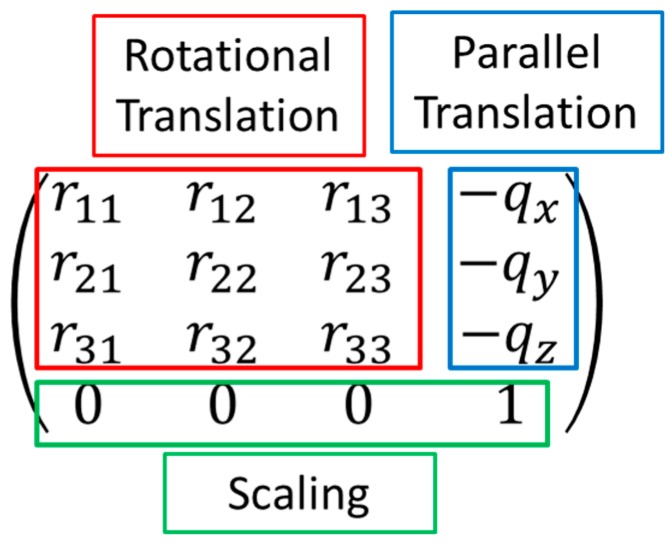
Translation matrix.

**Figure 7 sensors-19-01172-f007:**
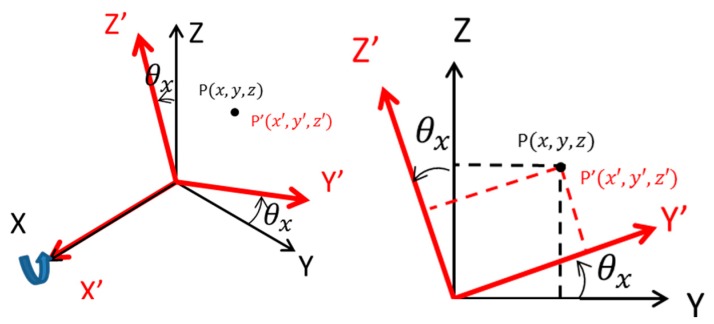
Rotational translation.

**Figure 8 sensors-19-01172-f008:**
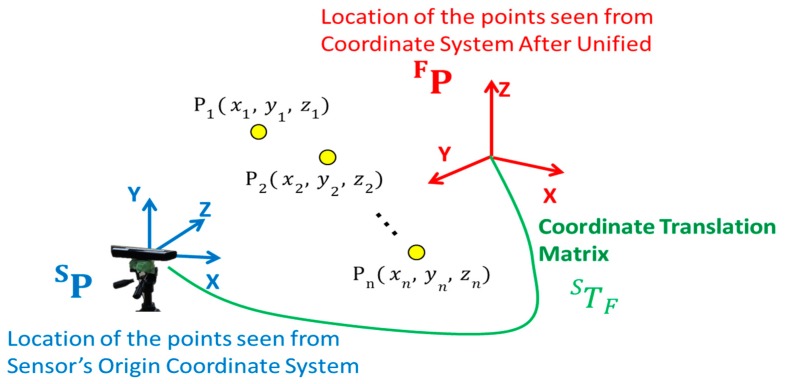
Coordinate system translation matrix.

**Figure 9 sensors-19-01172-f009:**
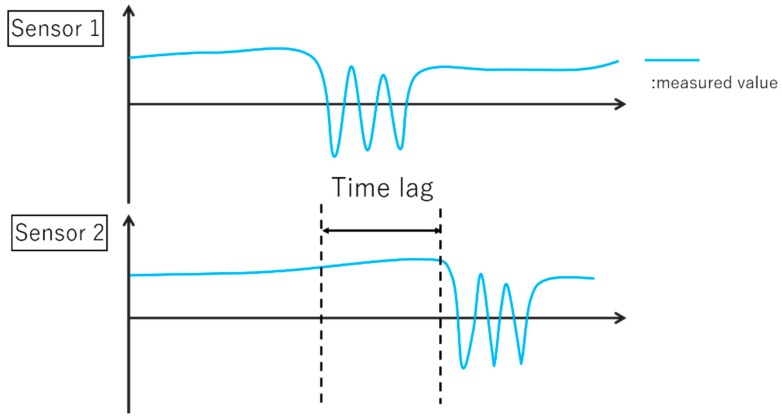
Time lag according to the difference of the time point of each sensor.

**Figure 10 sensors-19-01172-f010:**
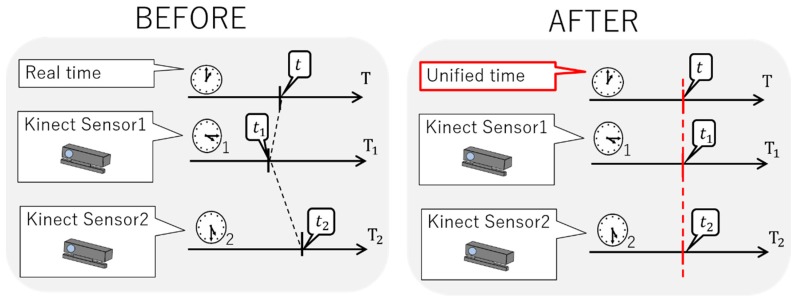
Time synchronization using unified time.

**Figure 11 sensors-19-01172-f011:**
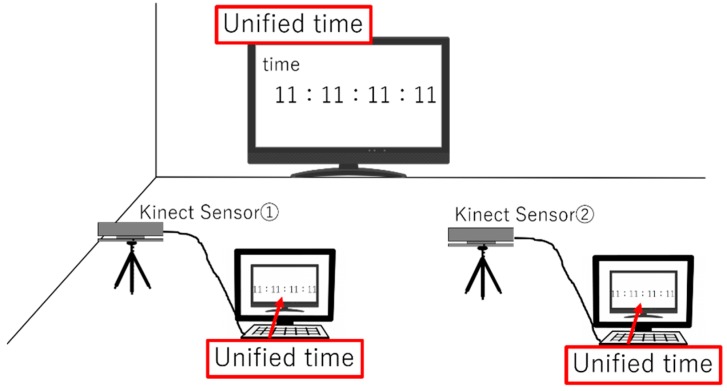
Time synchronization method.

**Figure 12 sensors-19-01172-f012:**
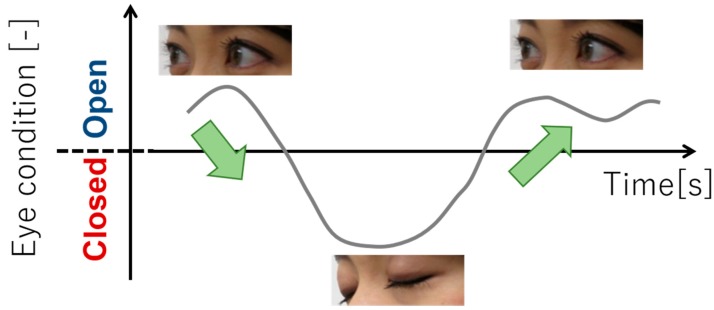
Recognition of eye condition.

**Figure 13 sensors-19-01172-f013:**
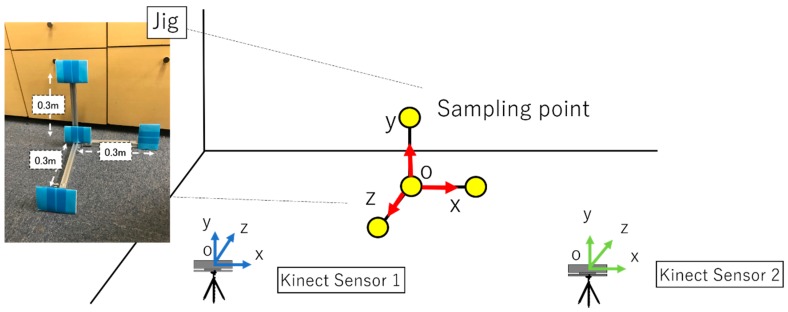
Experimental situation. The jig with four sampling points which are not on the same plane.

**Figure 14 sensors-19-01172-f014:**
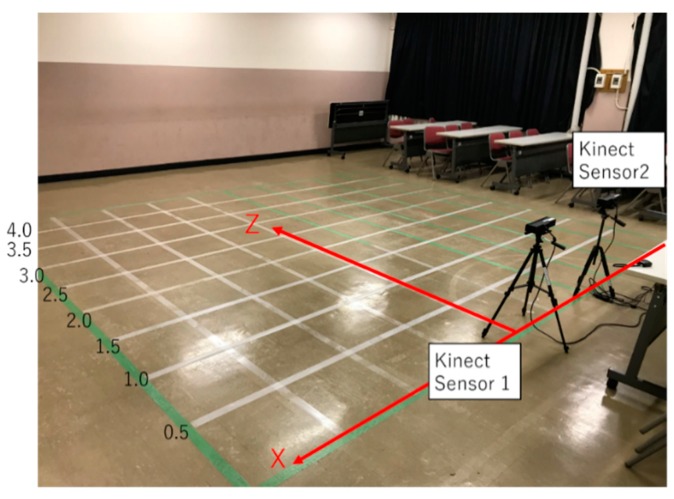
Experimental situation.

**Figure 15 sensors-19-01172-f015:**
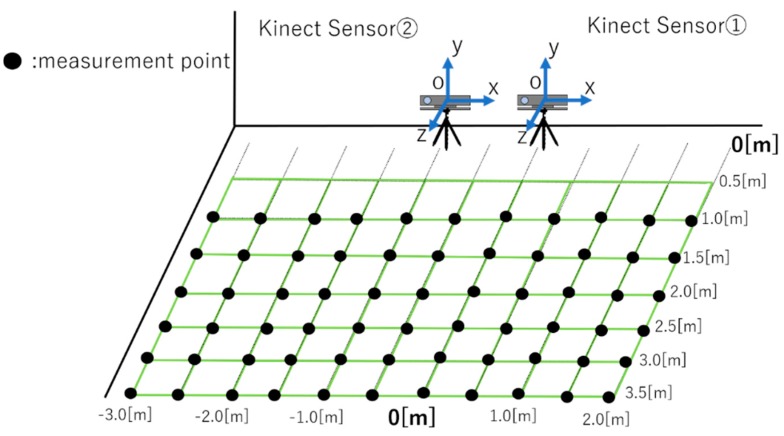
The total number of measurement points is 66 points.

**Figure 16 sensors-19-01172-f016:**
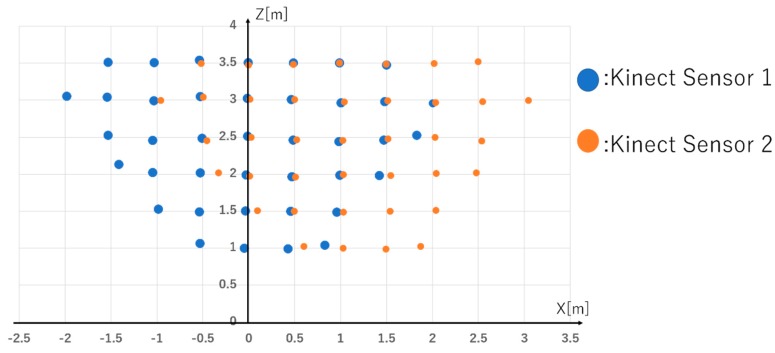
The total number of measurement points is 66 points.

**Figure 17 sensors-19-01172-f017:**
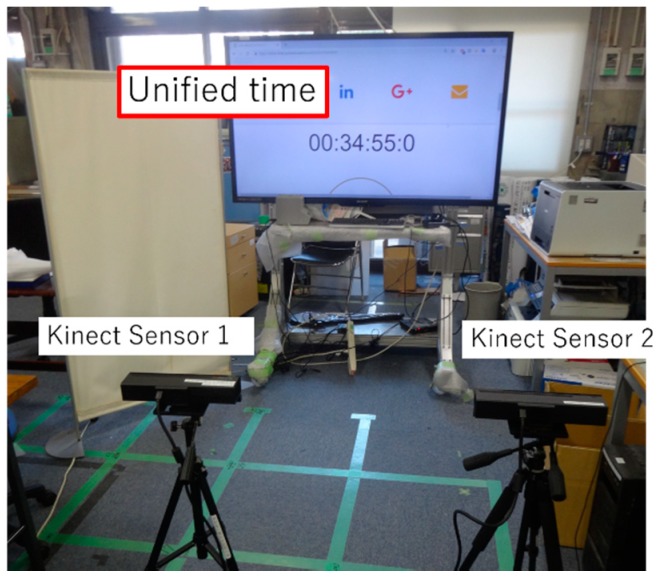
Experiment environment.

**Figure 18 sensors-19-01172-f018:**
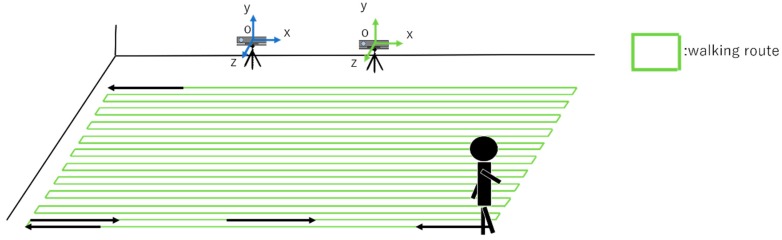
When multiple people overlap and the sensor cannot track multiple people.

**Figure 19 sensors-19-01172-f019:**
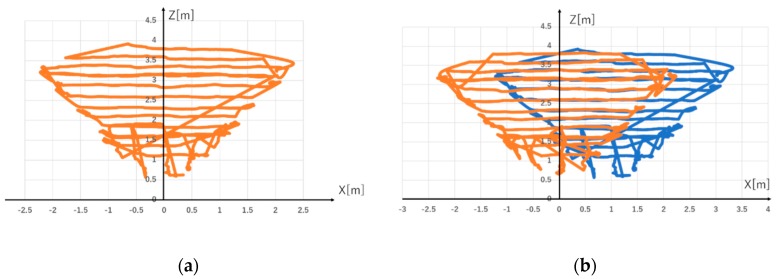
(**a**) shows that person tracking area of one Kinect Sensor. (**b**) shows that person tracking area of two Kinect Sensors.

**Figure 20 sensors-19-01172-f020:**
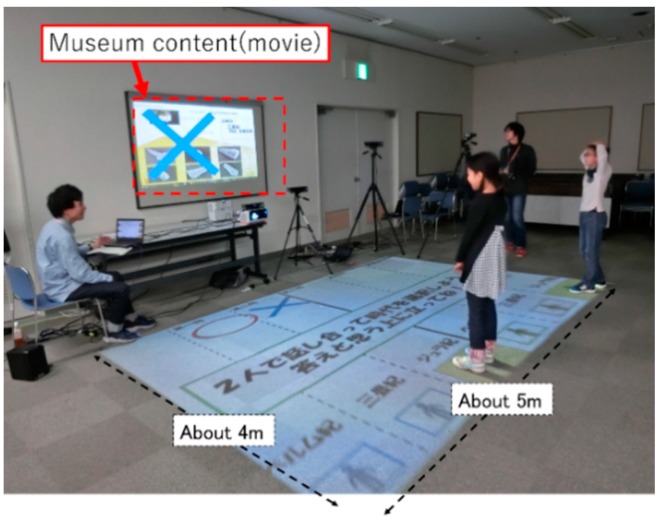
Evaluation experiment of museum contents [32]. We always sense four people within the measurement range.

**Figure 21 sensors-19-01172-f021:**
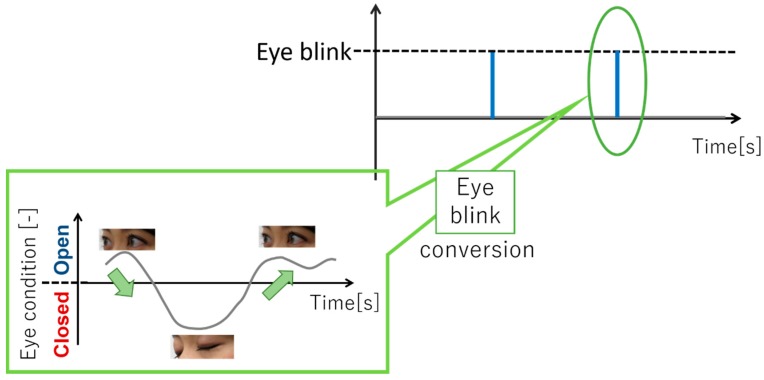
Eye blink conversion.

**Figure 22 sensors-19-01172-f022:**
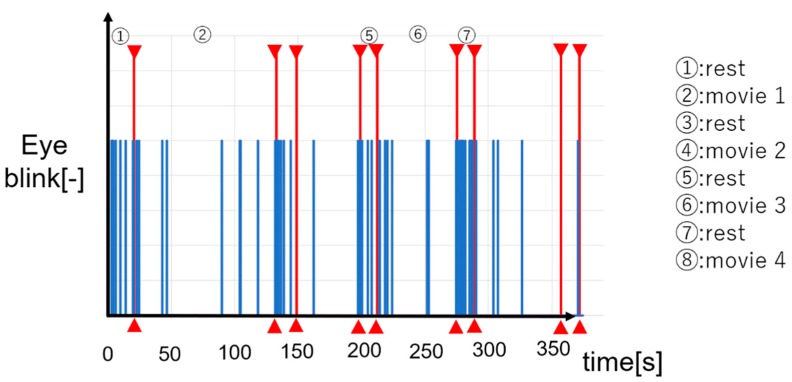
Experimental result after eye blink conversion.

**Table 1 sensors-19-01172-t001:** Error of the coordinates obtained by coordinate conversion of the coordinates measured by Kinect Sensor 2.

	X [cm]	−2.0	−1.5	−1.0	−0.5	0	0.5	1	1.5	2.0
Z [cm]										
**0**		-	-	-	-	-	-	-	-	-
**0.5**		-	-	-	-	-	-	-	-	-
**1**		-	-	-	10.35	2.47	1.31	1.31	-	-
**1.5**		-	-	9.35	0.489	2.98	3.78	3.67	-	-
**2.0**		-	16.92	2.74	3.87	2.98	4.52	4.02	8.09	-
**2.5**		-	6.25	2.64	4.07	4.83	2.84	2.48	5.20	-
**3.0**		4.10	4.03	1.78	1.00	4.55	1.21	4.57	6.19	4.36
**3.5**		-	2.03	2.38	2.22	1.52	1.28	1.56	2.09	-

**Table 2 sensors-19-01172-t002:** Experimental result of time synchronization.

	Start of Experiment		End of Experiment		Result	
	Unified Time [s]	Kinect Time [s]	Unified Time [s]	Kinect Time [s]	Elapsed Time (Unified Time) [s]	Elapsed Time (Kinect Time) [s]
**Kinect Sensor 1**	0	2.091	53.000	55.045	53.000	52.954
**Kinect Sensor 2**	0	3.979	53.000	57.012	53.000	53.033

**Table 3 sensors-19-01172-t003:** Eye blink rate in each section.

	1	2	3	4	5	6	7	8
**Eye blink rate**	19.92	5.37	23.48	3.75	19.48	11.80	54.40	3.69

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
