# Peer review of "Implementation and Evaluation of a Wide-Range Human-Sensing System Based on Cooperating Multiple Range Image Sensors"

_sensors, 2019, doi:10.3390/s19051172_

Round 1
Reviewer 1 Report
This paper aims to investigate the use of multiple Kinect sensors for quantifying the interest of museum visitors. The people's interest is estimated by measuring eye blink rates. The use of cooperating (spatially and temporally synchronized) Kinect sensors allows to enlarge the measurement range as well as to overcome problems of people overlapping.
Although the relevance of the topic, this reviewer has some issues with this paper. First, it lacks a lot of detail, the architecture is only described on a very high level, omitting all technical stuff like computer vision algorithms (e.g., people detection, segmentation and tracking) and all implementation details. Second, this reviewer does miss the actual contribution of this paper with respect to detecting eye blink with multiple Kinects: as the authors point out, they actually used well-known approaches in the sensor community, i.e. coordinate system transformation using a roto-translation matrix technique, a unified clock-based temporal synchronization, and a (not further detailed) eye blink detection.
To summarize, the lack of novelty in the methodological part as well as the low level of technical details, prevent this reviewer to agree with the publication of this paper.
Author Response
Response to Reviewer 1 Comments
Manuscript ID: sensors-411757
Dear Professor,
Many thanks for the constructive and encouraging comments on our manuscript from three reviewers. We enclose a carefully revised manuscript according to the comments and suggestions made. We provide an item-by-item response to all comments. The responses are start with “Response” which is included in “bold”. We have also made some non-requested minor typographical and readability edits.
We hope that these clarifications and revisions will now enable the paper to be accepted for publication in Sensors, and look forward to hearing from you soon.
Yours sincerely,
Mikihiro Tokuoka (on behalf of all authors)
Point 1: First, it lacks a lot of detail, the architecture is only described on a very high level, omitting all technical stuff like computer vision algorithms (e.g., people detection, segmentation and tracking) and all implementation details.
Response 1: We added all technical stuff like computer vision algorithms (eg, people detection, segmentation and tracking) and all implementation details to realize the system. We will explain below.
・coordinate transformation algorithm
First, we explain the algorithm of coordinate transformation.
① First, each Kinect sensor recognizes the coordinates of sampling points for calculating the coordinate transformation matrix.
② Irradiate infrared rays into the field of view of the sensor and measure the position coordinates of the specified sampling point from the reflection time, depth information and color image information.
③ On the basis of the coordinates measured by the Kinect sensor 1 and the coordinates measured by the Kinect sensor 2, the matrix conversion from matrix to expression is performed on all the PCs connected to the two Kinect sensors on the program to calculate the coordinate transformation matrix.
④ Based on the calculated coordinate transformation matrix, the coordinate system of the Kinect sensor 2 is unified to the coordinate system of the Kinect sensor 1.
Therefore, it is possible to think all the information on the coordinate axis of the Kinect sensor 1. As shown in Figure 4.1, the coordinate system is unified on the programmed computer.
|
Fig.4.1 When people overlap each other in the sensor’s line of sight, the sensor cannot track these people. The sensor cannot recognize person B hidden behind person A. |
We have added and modified several sentences as follows.
276 – 285 Next, we describe the method of processing persons and images programmed. First, each Kinect sensor recognizes the coordinates of sampling points for calculating the coordinate transformation matrix. Infrared rays were then irradiated into the field-of-view of the sensor and the position coordinates of the specified sampling point were measured according to the reflection time, depth information, and color-image information. Based on the coordinates measured using Kinect sensors 1 and 2, the matrix conversion from matrix to expression was performed on all the PCs connected to the two Kinect sensors on the program to calculate the coordinate transformation matrix. Based on the calculated coordinate transformation matrix, the coordinate system of Kinect sensor 2 was unified to the coordinate system of Kinect sensor 1. Therefore, it is possible to link all the information on the coordinate axis of Kinect sensor 1.
Response:
·Blinking eye measurement algorithm
Next we explain the algorithm of eye blink measurement.
Eye blink detection is a flow of human detection, skeleton information detection, face recognition, pupil detection. We explain the details. First, Kinect sensor performs human detection. Kinect sensor detect human using a database based on learning data of skeleton information of a large number of people. Next, we identifies person based on the coordinate information of human. Through these flow, we realize tracking of the same person in a multiple people environment. Then, it recognizes a human face and extracts the 3D coordinates of 1347 points on the face as feature points. Based on these information, we recognize the eye position. Next, we recognize eye blink. We identified the two states of the pupil (open eye / closed eyes), based on the ratio of the iris width to the maximum width of the iris obtained by counting the number of black pixels. We recognize "OPEN" when opening eyes and "CLOSE" when closing eyes based on the set threshold value.
We have added and modified several sentences as follows.
239 – 248 Eye-blink detection follows the flow of human detection, skeleton information detection, face recognition, and finally pupil detection, as detailed in the following text. First, the Kinect sensor performs human detection by using a database based on the learning data of skeletal information of a large number of people. Next, the sensor identifies a person based on the person’s coordinate information. Accordingly, the same person was tracked in a multipeople environment. Then, the sensor recognizes a human face and extracts the 3D coordinates of 1347 points on the face as feature points [31]. Based on these information, the eye position, and then the eye blinking were determined. The two states of the pupil (open/closed eyes) were determined based on the ratio of the iris width to the maximum iris width obtained by counting the number of black pixels. The state was recognized as "OPEN" when opening eyes and "CLOSE" when closing eyes based on the set threshold value.
Response:
·Time synchronization algorithm
Next we explain the algorithm of time synchronization.
A personal computer that connects Kinect Sensor 1 and Kinect Sensor 2, and records the unified time on the Internet by using the DataTime class in the program. Time on the Internet refers directly to the time of the existing NTP server, and in this research we referred to the NTP server of Tokyo Science University (nodarntp.rs.noda.tus.ac.jp) that can be used within our university. The time of the Internet was recorded at 30 fps which is the sampling rate of Kinect Sensor.
However, depending on the performance of the personal computer, the recorded time of the time differs by approximately 1 ms. Therefore, the error between the time of the NTP server and the time of the recording was constantly calculated in each personal computer, and correction was made. As a result, time synchronization was performed.
We have added and modified several sentences as follows.
216 – 225 Next, we explain the algorithm of time synchronization. A personal computer was used to connect Kinect Sensors 1 and 2 and record the unified time on the Internet by using the DataTime class in the program. Time on the Internet refers directly to the time of the existing NTP server, and in this research, we referred to the NTP server of Tokyo Science University (nodarntp.rs.noda.tus.ac.jp); it can be used within our university. The time of the Internet was recorded at 30 fps, which is the sampling rate of a Kinect Sensor. However, depending on the performance of the personal computer, the recorded time differs from the unified time by approximately 1 ms. Therefore, the error between the times of the NTP server and recording was constantly calculated in each personal computer, and a correction was made. As a result, time synchronization was performed.
These algorithms, that is, the program is running.
Point 2: Second, this reviewer does miss the actual contribution of this paper with respect to detecting eye blink with multiple Kinects: as the authors point out, they actually used well-known approaches in the sensor community, i.e. coordinate system transformation using a roto-translation matrix technique, a unified clock-based temporal synchronization, and a (not further detailed) eye blink detection.
Response 2: We explain the actual contribution of this paper with respect to detecting eye blink with multiple Kinect.
We focused on eye blink to quantify the interest of multiple people at the museum. Clearly, eye blink is suppressed when people's interests and attention work. In other words, when people are more interested, the number of their eye blink decreases. If we measure eye blink at all times, we quantitatively estimate the interest, engagement, and excitement of "on the spot, at the time". However, in the conventional eye blink measurement method, the range of measurement is narrow, when learners look away, part of eye blink data is missing. If we realize the proposed wide-range eye blink measurement method, eye blink data is not missing when learners look away, because we measure eye blink with cooperated sensor. In other words, we always measure and record eye blink. By measuring interest quantitatively, we clarify the learners’ interest in the contents of the museums.
Below we will explain examples of contribution. There are times when you are experiencing content as shown in the photo on the Figure (a) and when you are watching the exhibits as shown in the photo on the Figure (b).
It is possible to compare the learning effect by calculating the blink rate of the visitor in each case when viewing the Figure (a) and viewing the Figure (b). Based on the result, it is possible to quantitatively determine which content is effective for the visitor. It is a great contribution to understand content that is effective for visitors and contents that are not so effective for visitors.
Fig. (a) Fig. (b)
We have added and modified several sentences as follows.
228 – 238 We focused on the eye blink to quantify the interest of multiple people at the museum. Clearly, eye blinking is suppressed when an entity has caught people's interests. In other words, when people are more interested, the number of their eye blinks decreases. If we measure eye blink at all times, we quantitatively estimate the interest, engagement, and excitement "on the spot, at the time". However, in the conventional eye-blink measurement method, the measurement range is narrow because when learners look away, part of the eye-blink data is missed. The realization of the proposed wide-range eye-blink-measurement method could capture the eye-blink even when learners look away because its measures eye blinks using a coordinated sensor. In other words, the sensor always measures and records eye blinks. By measuring interest quantitatively, we clarified the learners’ interest in the contents of the museums.
Response:
・We explain about novelty.
There are few studies using our proposed method in the coordinate transformation matrix. However, conventional research has not mentioned actual accuracy using Kinect V2 sensor. I think that it is very meaningful to evaluate the accuracy of the unification of the coordinate system using the coordinate transformation matrix of the Kinect V2 sensor.
We have added and modified several sentences as follows.
308 – 310 The average error value of the coordinate transformation was 4.18 [cm], and the standard deviation was ± 2.98 [cm]. Thus, coordinate transformation by using the Kinect sensor can unify the coordinate system with an accuracy of 4.18 ± 2.98 [cm].
Response:
Conventional time synchronization research is based on unified time and server established. Studies using unified time are not stated about the accuracy of time synchronization, assuming that the cycle of unified time does not fluctuate. However, there is a possibility that the unified time period is deviated.
Meanwhile, research that establishes a server and synchronizes time is accurate. However, in order to process and synchronize information with high capacity and geometric quality such as Kinect V2 sensor with color images and depth information, it is necessary to establish an expensive server with fast processing time. Establishing an expensive server has no versatility when we consider the realization at our museum which is our original purpose. Therefore, we did not set up a server, we referenced the existing server, time synchronization was done taking account of the processing time of each personal computer, and the accuracy was evaluated. It was clarified that this proposal can be performed with accuracy of 0.0069[s] without establishing an expensive server and time synchronization.
We have added and modified several sentences as follows.
197 – 207 Conventional time-synchronization research is based on the establishment of a unified time and server. Previous studies that used a unified time did not state about the accuracy of time synchronization, assuming that the cycle of unified time does not fluctuate. However, there is a possibility that the unified time deviates. In addition, any research that establishes a server and synchronizes time is accurate. However, to process and synchronize information with high capacity and geometric quality like the Kinect V2 sensor with color images and depth information, it is necessary to establish an expensive server with a fast processing time. The establishment of an expensive server has no versatility if we consider the realization at our museum; this is the original purpose of our study. Therefore, we did not set up a server, and referenced the existing server, conducted time synchronization by considering the processing time of each personal computer, and evaluated the accuracy.
334 – 336 The average value of time synchronization error was 0.0069 [s], and the standard deviation was ± 0.0024 [s]. As a result, time synchronization using the NTP server can be synchronized with an accuracy of 0.0069 ± 0.0024 [s].
Response:
The purpose of our research is the learning effect of contents and learning materials in the museum, that is, the quantification of interests of learners. In the actual museum, evaluation of the learning effect using our proposed technique has not been done at all. Questions such as questionnaires and interviews are not quantitative.
In the first place, museums are very important for children's science education. My research is to make learning more efficient at educational sites such as important museums, and this research is very meaningful.
We have added and modified several sentences as follows.
62 – 63 The learning effect in an actual museum has not been evaluated using our proposed technique.

Reviewer 2 Report
line 34 - VTR --> video tape recorder
84-85 - "The data storage [...] measurement." This sentence is not clear.
103 - subsection 2.2, not 2.1
Figure 8 is cut, pleas fix it.
line 177 - what is N=4?
188-89 - too many repetitions
193-94 - too many repetitions
198 - why you haven't described what is the "conventional synchronization of multiple Kinect sensors"? And please explain why it failed.
203-04 - please rewrite this sentence, trying to avoid repetitions.
207-08 - how much accurate is the accurate time synchronization that you ensure?
211 - gaged -> gauged ?
212 - 1300 points on the face. I think that it is too much, I can't find a reference.
subsection 3.1.2 - the font is smaller compared to other subsections, please fix it.
line 294 - what is the meaning of walk before?
Figure 21 is not clear, and maybe could be omitted.
Author Response
Response to Reviewer 2 Comments
Manuscript ID: sensors-411757
Dear Professor,
Many thanks for the constructive and encouraging comments on our manuscript from three reviewers. We enclose a carefully revised manuscript according to the comments and suggestions made. We provide an item-by-item response to all comments. The responses are start with “Response” which is included in “bold”. We have also made some non-requested minor typographical and readability edits.
We hope that these clarifications and revisions will now enable the paper to be accepted for publication in Sensors, and look forward to hearing from you soon.
Yours sincerely,
Mikihiro Tokuoka (on behalf of all authors)
Point 1: line 34 - VTR --> video tape recorder
Response 1: Done. We have revised the typos.
line 34 - Moreover, a panel and video tape recorder of explanation is used to complement the exhibits.
Point 2: 84-85 - "The data storage [...] measurement." This sentence is not clear.
Response 2: Done. We have revised the explanation as follows.
85-86 - The data storage unit includes elements, such as the direction of the learner’s face, detection of blinking, and gaze time in the gaze direction, related to the learner’s interest.
Point 3: 103 - subsection 2.2, not 2.1
Response 3: Done. We have revised the typos. And we have also checked whole article to make sure there is no similar mistakes any more.
Point 4: Figure 8 is cut, pleas fix it.
Response 4: Done. We cut out Figure 8 from the text. In addition, we corrected the figure number accompanying it.
Point 5: line 177 - what is N=4?
Response 5: We have revised the typos. It means not N=4 but n=4. Also, n stands for n in Equation 14. When n = 4, the matrix is a square matrix.
We have revised the explanation as follows.
174-176 - When equation (14) is n = 4 and these points are not on the same plane, the coordinate transformation matrix can be obtained as follows, using the inverse matrix of [28].
Point 6: 188-89 - too many repetitions
Response 6: Done. We have deleted the repetition. We have revised the explanation as follows.
186-188 - Through these calculations, it becomes possible to treat the three-dimensional coordinate measurement result measured by all Kinects as a unified value in the coordinate system of the origin Kinect. All these matrix calculations are performed on a program.
Point 7: 193-94 - too many repetitions
Response 7: Done. We have deleted the repetition. We have revised the explanation as follows.
191-192 - In the measurement by multiple sensors, the measurement start timing and the end timing is different, because the time of each sensor is different.
Point 8: 198 - why you haven't described what is the "conventional synchronization of multiple Kinect sensors"? And please explain why it failed.
Response 8: Regarding reference [29], time synchronization is not mentioned even though time synchronization is necessary. On the other hand, reference [30] mentions that it is establishing a time synchronization server on its own and time synchronization. However, when actually tracking a person and processing personal behavior information (measurement of position coordinates of plural people, eye blink measurement), a lot of information is transmitted to the server. Then, in theory, time synchronization can be done, but in reality the cycle will always fluctuate and time synchronization can’t be done. Therefore, general versatility is low and it can’t be said that synchronization has been achieved.
We have revised the explanation as follows.
197 – 206 Conventional time-synchronization research is based on the establishment of a unified time and server. Previous studies that used a unified time did not state about the accuracy of time synchronization, assuming that the cycle of unified time does not fluctuate. However, there is a possibility that the unified time deviates. In addition, any research that establishes a server and synchronizes time is accurate. However, to process and synchronize information with high capacity and geometric quality like the Kinect V2 sensor with color images and depth information, it is necessary to establish an expensive server with a fast processing time. The establishment of an expensive server has no versatility if we consider the realization at our museum; this is the original purpose of our study. Therefore, we did not set up a server, and referenced the existing server, conducted time synchronization by considering the processing time of each personal computer, and evaluated the accuracy.
Point 9: 203-04 - please rewrite this sentence, trying to avoid repetitions.
Response 9: Done. We have deleted the repetition. We have revised the explanation as follows.
211-212 - First, we created unified time and recorded the unified time on multiple sensors; then, we unified the time of multiple sensors to this unified time and realized time synchronization.
Point 10: 207-08 - how much accurate is the accurate time synchronization that you ensure?
Response 10: We need to know when the eye blink has occurred. Eye blink happens once at 0.20 [s]. For this reason, we set 0.20 [s] as the tolerance for the accuracy of time synchronization obtained in this research. The accuracy of time synchronization secured by us adds sentences to 3.2.2. Also, we deleted the 207-08 sentences.
We will ensure accuracy of 0.0069 ± 0.0024 [s].So we added the text to the following part.
334-336 - The average value of time synchronization error was 0.0069 [s], and the standard deviation was ± 0.0024 [s]. As a result, time synchronization using the NTP server can be synchronized with an accuracy of 0.0069 ± 0.0024 [s].
Point 11: 211 - gaged -> gauged ?
Response 11: Done. We have revised the typos. We have revised the explanation as follows.
228 - It is said that people's interest can be gauged using their eye blink.
Point 12: 212 - 1300 points on the face. I think that it is too much, I can't find a reference.
Response 12: The Kinect sensor finds 1347 feature points of the face. Next, based on that information, the Kinect sensor can measure by estimating the position of both eyes, the position of the nose and the position of the mouth. We write a reference of 1347 points below.
Cafaro, A.; Wagner, J.; Baur, T.; Dermouche, S.; Torres Torres, M.; Pelachaud, C.; Valstar, M. The NoXi database: multimodal recordings of mediated novice-expert interactions. In Proceedings of the 19th ACM International Conference on Multimodal Interaction, 2017, pp. 350-359. ACM.
We added a reference and fixed the number associated with it.
We have revised the explanation as follows.
243-244 – Then, it recognizes a human face and extracts the 3D coordinates of 1347 points on the face as feature points[31].
Point 13: subsection 3.1.2 - the font is smaller compared to other subsections, please fix it.
Response 13: Done. We have revised the typos. We have revised the same 13px as other subsections. And we have also checked whole article to make sure there is no similar mistakes any more.
Point 14: line 294 - what is the meaning of walk before?
Response 14: Done. We have revised the typos. That means not “before” but “in front of”. We have revised the explanation as follows.
346 - In the experiment, the subject walks in front of two Kinect Sensors.
Point 15: Figure 21 is not clear, and maybe could be omitted.
Response 15: Done. Thanks for your advice. It is much clearer than before. The figure shows that converting the action of a single eye blink to a single pulse signal on the graph. We have revised the figure as follows.
Figure 20. Eye blink conversion.

Round 2
Reviewer 1 Report
This paper aims to investigate the use of multiple Kinect sensors for quantifying the interest of museum visitors. The use of cooperating (spatially and temporally synchronized) Kinect sensors allows to enlarge the measurement range as well as to overcome problems of people overlapping.
In this version of the manuscript, the authors carried out a careful editing of English language and style, as well as they added some qualitative details to Section 2.3 "Time Synchronization" and to Section 2.4 "Eye Blink Detection". However, the added details refer mainly to the use of time synchronization via an NTP server and of computer vision algorithms already provided by the Kinect framework (e.g., human detection, skeleton detection, face recognition, eye detection, etc.). This cannot be considered an original scientific contribution. Thus, in the opinion of this reviewer, the manuscript is not worthy of publication.
Author Response
Response to Reviewer 1 Comments
Manuscript ID: sensors-427850
Dear Professor,
Many thanks for the constructive and encouraging comments on our manuscript. We enclose a carefully revised manuscript according to the comments and suggestions made. We provide an item-by-item response to all comments. The responses are start with “Response” which is included in “bold”. We have also made some non-requested minor typographical and readability edits.
We hope that these clarifications and revisions will now enable the paper to be accepted for publication in Sensors, and look forward to hearing from you soon.
Yours sincerely,
Mikihiro Tokuoka (on behalf of all authors)
Point 1: However, the added details refer mainly to the use of time synchronization via an NTP server and of computer vision algorithms already provided by the Kinect framework (e.g., human detection, skeleton detection, face recognition, eye detection, etc.). This cannot be considered an original scientific contribution.
Response 1: There is no research that the system combining all the functions of time synchronization via an NTP server and of computer vision algorithms (eg, human detection, skeleton detection, face recognition, eye detection, etc.) is operating. Our research is that this system is operating in the museum. This system works on one program. These things become scientific contributions.
